# Chelators for Treatment of Iron and Copper Overload: Shift from Low-Molecular-Weight Compounds to Polymers

**DOI:** 10.3390/polym13223969

**Published:** 2021-11-17

**Authors:** Martin Hruby, Irma Ivette Santana Martínez, Holger Stephan, Pavla Pouckova, Jiri Benes, Petr Stepanek

**Affiliations:** 1Institute of Macromolecular Chemistry, Academy of Sciences of the Czech Republic Heyrovského Náměstí 2, 162 06 Prague, Czech Republic; stepan@imc.cas.cz; 2Helmholtz-Zentrum Dresden-Rossendorf, Institute of Radiopharmaceutical Cancer Research Bautzner Landstraße 400, 01328 Dresden, Germany; santan75@hzdr.de (I.I.S.M.); h.stephan@hzdr.de (H.S.); 3Institute of Biophysics and Informatics, First Faculty of Medicine, Charles University in Prague, Salmovska 1, 120 00 Prague, Czech Republic; pavla.pouckova@lf1.cuni.cz (P.P.); jiri.benes@lf1.cuni.cz (J.B.)

**Keywords:** iron, copper, polymer, chelator, Wilson’s disease, hemochromatosis

## Abstract

Iron and copper are essential micronutrients needed for the proper function of every cell. However, in excessive amounts, these elements are toxic, as they may cause oxidative stress, resulting in damage to the liver and other organs. This may happen due to poisoning, as a side effect of thalassemia infusion therapy or due to hereditary diseases hemochromatosis or Wilson’s disease. The current golden standard of therapy of iron and copper overload is the use of low-molecular-weight chelators of these elements. However, these agents suffer from severe side effects, are often expensive and possess unfavorable pharmacokinetics, thus limiting the usability of such therapy. The emerging concepts are polymer-supported iron- and copper-chelating therapeutics, either for parenteral or oral use, which shows vivid potential to keep the therapeutic efficacy of low-molecular-weight agents, while avoiding their drawbacks, especially their side effects. Critical evaluation of this new perspective polymer approach is the purpose of this review article.

## 1. Introduction

The essential elements iron and copper are needed for the proper function of redox processes in every organism, and their metabolism is strictly regulated [1,2,3,4]. Deficiencies of iron and copper are common and are reviewed elsewhere [5,6]. However, the excess of iron and copper is dangerous, as these elements as free ions catalyze the formation of reactive oxygen species (ROS), which may damage the organism in an even fatal way if untreated. The main organ damaged is always the liver, as it is the key storage organ for both iron and copper, but also more specific organ damages (e.g., of neural system for copper or of gonads for iron) occur (for a more detailed discussion, see Section 3 and Section 4). The current state of the art iron and copper overload therapies mostly rely on low-molecular-weight chelators. As these low-molecular-weight metal chelators possess strong side effects (e.g., ophthalmic and auditory toxicity; pulmonary, renal and neurological effects for deferoxamine or gastrointestinal problems; leucopenia and liver problems for penicillamine; for a more detailed discussion, see Section 3 and Section 4), new treatment paradigms are to be discovered, for which polymers hold great promise. The purpose of this review article is to critically compare the advantages and disadvantages of currently developed polymer-based treatment approaches for iron and copper overload with a special focus on the correlation structure–properties–function–observed advantages/disadvantages and to propose new ways ahead.

## 2. Coordination Chemistry of Iron and Copper

The design of iron- and copper-chelating agents was initially based on relatively simple structures of complexing agents [7,8,9,10]. Essentially, principles of classical coordination chemistry, which goes back to Alfred Werner, have been used for this purpose [11]. This has led to a series of small chelating ligands for sequestering iron and copper ions, which, with a better understanding of the coordination and pharmacokinetic properties, as well as improvement of the art of synthesis, have led to metal–ligand complexes that have both a matched thermodynamic stability and a desired kinetic inertness. In the following, classical complexing agents, as well as new structures, for the stable and selective binding of iron and copper ions are shown and briefly discussed.

Iron and copper are transition metals of the fourth period. With regard to the coordination behavior, there are clear differences for these two elements, mainly coming from different preferred coordination geometry, size, charge and polarizability of their ions (see below). This is due in particular to the different electronic and redox properties of these ions. Iron ions occur in biological systems mainly in the oxidation states 2+ (d^6^ configuration) and 3+ (d^5^), which are stabilized by different ligands according to Pearson’s hard and soft (Lewis) acids and bases (HSAB) concept [12]. This concept deems species that are small, highly charged and weakly polarizable as “hard” and species that are large, lowly charged and highly polarizable as “soft”. “Hard” Lewis acids prefer “hard” Lewis bases and vice versa. While the hard Lewis acid Fe(III) forms coordination compounds preferably with hard Lewis bases, such as oxygen-containing ligands (hydroxide, phenolate and carboxylate), Fe(II) tends to prefer nitrogen ligands, such as amines and imidazole, due to lower charge density. Under aerobic conditions, mainly Fe(III) is found in biological systems. The coordination chemistry of iron is dominated by hexacoordinate complexes with an octahedral coordination geometry. This can be achieved by complexation with six mono-, three bi-, two tri- or one hexadentate, or by the combination of ligands with different denticities (Figure 1).

Multidentate ligands are favored due to a higher stability, which is achieved by a chelating effect [13]. The search for ideal chelating agents for the treatment of systemic iron overload has been the subject of intensive research for years [14,15,16]. With a radius of 0.65 Å, Fe(III) is a small cation. To achieve saturation of the coordination sphere, small sterically undemanding ligands are required. These include, in particular, bidentate ligands, such as catechols, hydroxamates, 8-hydoxyquinolines (8HQ) and hydroxypyridinones (HOPO). Due to their high flexibility, multidentate aminocarboxylates, such as EDTA, are also suitable for the formation of hexacoordinate complexes with Fe(III) (Figure 2). The hexadentate iron chelator Deferoxamine DFO (vide infra), consisting of four hydroxamate groups, also forms hexacoordinate Fe(III) complexes.

With regard to their use in chelation therapy for iron, derivatives of 8-hydoxyquinoline and hydroxypyridinone should be discussed in particular. For instance, 8HQ forms very stable 3:1 complexes with Fe(III) (log K_1_ = 13.7, log β_3_ = 37) [17]. Due to the redox lability of these complexes, in vivo use is not harmless [18]. Clioquinol, PBT2, HLA20, M30 and VK-28 have been the subject of clinical trials for the treatment of neurogenerative diseases where sequestering of Fe(III) is required [19,20,21,22]. However, due to a rather low chelation selectivity of these 8-hydroxyquinoline derivatives (clioquinol, PBT2, HLA20, M30 and VK-28) for iron(III) ions, these derivatives may exhibit in vivo toxicity. This is because they may chelate, e.g., copper(II) or zinc(II) along with iron(III), causing depletion of these essential micronutrients and further leading to inhibition of metalloenzymes dependent on them [23]. HOPO ligands also bind Fe(III) very strongly. Of the three positional isomers, 3-hydoxypyridin-4-one forms Fe(III) complexes of the highest stability (log K_1_ = 14.2, log β_3_ = 37.2) [24]. This structural element can be incorporated into a wide range of ligand structures of different denticities [25]. The overall favorable complexation properties of the bidentate 3-hydoxypyridin-4-one ligand have led to the development of Deferiprone (1,2-dimethyl-3-hydroxypyridin-4-one) for clinical use in iron detoxification [14]. Deferasirox is an iron chelating agent based on 1,2,4-triazol with three appending aromatic rings that forms a 2:1 complex with Fe(III) having a distorted octahedral structure. The Fe(III) complex of deferasirox has very high stability and is hydrophobic in character, enabling to chelate iron from hepatocellular stores [26,27]. The DFO forms a hexacoordinate complex with Fe(III) with distorted octahedral geometry [28]. In terms of thermodynamic stability and kinetic inertness, DFO embodies ideal properties for clinical use as a Fe(III) chelator.

Due to its electron configuration, copper can assume the oxidation states 1+ to 4+, but the oxidation numbers 1+ and 2+ dominate in stable copper complex compounds. The d^10^ configured Cu(I) forms diamagnetic complex compounds, whereby the tetrahedral geometry is favored. Cu(I) prefers ligands based on soft Lewis bases with donor groups such as isocyanides, sulfur in thioethers (methionine), phosphorus in phosphanes or also thiolates (cysteine). The rather low stability of many Cu(I) complexes under physiological conditions largely precludes in vivo applications [29].

The Cu(II) can be classified as a “borderline” (something between “hard” and “soft” Lewis acid) element according to Pearson’s HSAB concept. The Cu(II) complexes have coordination numbers four, five and six and usually favor nitrogen-containing ligands such as amines, but especially heteroaromatics such as pyridine, pyridazine and imidazole. The d^9^ configuration in the paramagnetic Cu(II) complexes is often characterized by a tetragonal distortion of the coordination geometry due to the Jahn-Teller effect [30]. Consequently, an octahedral geometry results in an elongation of axial ligands, which in extreme cases leads to the loss of ligands along the *z*-axis and thus to a square-planar or square-pyramidal structure of the corresponding Cu(II) complexes. Overall, the copper(II) coordination chemistry shows a high diversity and includes tetracoordinate square-planar, pentacoordinate square-pyramidal and trigonal-bipyramidal, as well as hexacoordinate octahedral and trigonal-prismatic, complexes (Figure 3).

A whole range of different complexing agents is available for the stable complex formation of Cu(II) in vivo (Figure 4).

A large variety of amine-based polydentate chelating agents have been investigated, such as polyaminocarboxylates (ethylenediaminetetraacetic acid (EDTA) and diethylenetriaminepentaacetic acid (DTPA)) [31,32]; cyclic polyamines (azamacrocycles TACN, cyclen and cyclam) [33,34,35]; cyclic polyaminocarboxylates (1,4,7-triazacyclononane-1,4,7-triacetic acid (NOTA), 1,4,7,10-tetraazacyclododecane-1,4,7,10-tetraacetic acid (DOTA) and 1,4,8,11-tetraazacyclotetradecane-1,4,8,11-tetraacetic acid (TETA)) [36,37,38]; and cage compounds, such as Sargeson’s sarcophagine systems (sar-type) [39,40]. Due to the high complementarity of the electronic properties of Cu(II) with the pyridyl group, pyridine compounds play a prominent role in the development of Cu(II) complexes of high stability and selectivity [41]. These include both open-chain and macrocyclic pyridine-containing polyamines (*N*,*N*′,*N*′′-tris(2-pyridylmethyl)-*cis*,*cis*-1,3,5-triaminocyclohexane—tachpyr, 2-[4,7-bis(2-pyridylmethyl)-1,4,7-triazacyclononan-1-yl]acetic acid—DMPTACN-COOH) [34,42,43,44]. The ligand developments have been significantly advanced with a view to use in radiopharmacy [45,46,47]. Copper complexes of very high kinetic inertness are required here, and complex formation may require harsh conditions. This applies, for example, to crossbridged macrocyclic compounds such as tetraazabicyclo[6.6.2]hexadecane-4,11-diacetic acid—CB-TE2A [48,49]. For use in chelation therapy, however, there are other requirements. Here, stable and selective complex formation of the metal ion under physiological conditions is important and must take place very quickly. The complex formed must then be excreted from the body. This means that open-chain compounds, in particular, are in demand. There are a number of new developments in this field. This concerns, in particular, open-chain ligands that are very rigid, optimally pre-organized and complementary to Cu(II). These include pyridine-containing bispidine (3,7-diazabicyclo[3.3.1]nonane) ligands that form highly stable Cu(II) hexacoordinate complexes of distorted octahedral geometry under mild conditions with high complex formation kinetics [50,51,52,53]. The metal ion selectivity can be varied by variation of the donor groups [54]. Very recently, the ligand H_2_pyhox based on an ethylenediamine backbone and containing pyridine and 8-hydroxyquinoline groups was developed [55]. This ligand has high rigidity and optimal pre-organization for Cu(II) and forms a hexacoordinate complex of very high thermodynamic stability (log *K*_Cu-L_ = 26.63) under physiological conditions, which also exhibits high kinetic inertness in vivo.

A variety of ligands have been optimized for complexation of Cu(II); in particular, triethylenetetramine (trien), tetrathiomolybdate ([MoS4]^2−^]) and penicillamine have been used for copper chelation therapy [9,23]. Trien forms 1:1 complex with Cu(II), which shows high thermodynamic stability (log K_Cu-L_ = 20.3). The complex has an octahedral coordination geometry, whereby the four nitrogen atoms of the ligand are arranged in an equatorial plane and the remaining two positions are occupied by anions or water [56,57]. Tetrathiomolybdate and penicillamine form polynuclear cluster structures with copper ions [58,59].

The chelator types are summarized in Table 1 for both iron and copper. In general, complexing agents for chelation therapy must possess even optimal complexing properties, favorable pharmacokinetics and biodistribution behavior, as well as a reasonable elimination rate from the body.

## 3. Iron Uptake and Metabolism, Diseases Connected with Iron Overload and Current Treatment Strategies

Iron is an essential element needed for many redox processes of the cell. Most iron in the diet is in the form of iron(III), weakly or more strongly bound to organic ligands present in the biological matrix [60]. During digestion, iron(III) is released, and because it is practically not absorbable, it must be reduced by intestinal oxidoreductase to iron(II) [61]. Iron(II) is then uptaken into intestinal mucosa cells by divalent metal transporter 1 (DMT1) [61]. Meat contains mainly heme-bound iron, which is the most bioavailable form of iron in the diet [60]. Heme-bound iron uptake is different; after digestion of protein part of the hemoproteins, heme is uptaken into intestinal mucosa cells by a dedicated heme transporter mechanism, and then iron is decomplexed intracellularly by oxidation with heme oxygenase [62,63].

Compounds in food that themselves reduce iron(III) to iron(II) (e.g., ascorbate or lactate) promote iron absorption [64,65], while compounds complexing iron (e.g., phytin or tannins; see below) [60] decrease its bioavailability. Dietary beta carotene significantly increases the bioavailability of iron [66,67] by means of a not yet fully understood process.

Hemochromatosis is a group of hereditary recessive genetic disorders characterized by the toxic accumulation of iron in parenchymal organs with normal iron-driven erythropoiesis. It can be caused by mutations in any gene that limits iron entry into the blood, resulting in unregulated excessive iron uptake into the body [68]. Five major categories of hereditary hemochromatosis have been described according to different types of mutations [69,70]: (i) high Fe-related hemochromatosis (type 1, variation in HFE gene, which is responsible for regulation of the intestinal uptake and biodistribution of iron) [71], (ii) juvenile hemochromatosis (type 2A and 2B) [72], (iii) transferrin receptor 2 hemochromatosis (type 3) [69], (iv) ferroportin disease (type 4A and 4B) [73] and (v) aceruloplasminemia/hypoceruloplasminemia [74]. Worldwide prevalence of the most common type 1 is 1:200–400 (depending on the population studied), and clinical presentation usually occurs in middle age [74,75]. Progressive accumulation of iron occurs in the liver and in many other tissues, including the pancreas, skin, heart, joints and the gonadotrophin-secreting cells of the pituitary. When reaching toxic iron levels, i.e., after depletion of protective mechanisms, these organs become oxidatively damaged by reactive oxygen species produced by the Fenton reaction catalyzed by iron. This leads to hepatic fibrosis, diabetes mellitus, arthropathy, pigmentation, cardiomyopathy and hypogonadotropic hypogonadism [70,74]. Serum ferritin levels and transferrin saturation are typically raised in presence of the disease [70,71]. The two most common complications of the disease that can affect prognosis and survival are cirrhosis and hepatocellular carcinoma [71,76].

Homeostatic balance requires only 1–3 mg of absorbed iron per day to replenish losses from desquamated cells. Higher iron losses may occur physiologically (due to menstrual bleeding or higher consumption during pregnancy of women) or generally due to any injury or pathological bleeding. It is very important that there are no regulated means of iron excretion, and therefore dietary iron absorption (primarily from duodenal enterocytes) is to be highly regulated [60]. Reticuloendothelial cells serve as the major hepcidin-regulated iron repository. They obtain most of their iron from the phagocytosis of senescent erythrocytes. At equilibrium, these cells release about 25 mg of iron each day. Similar to reticuloendothelial cells, hepatocytes have the importance of iron storage in the form of ferritin. Most importantly, they have a central role in iron homeostasis as the site of production of the hormone hepcidin. Hepcidin production is regulated by iron status, erythropoietic activity, oxygen tension and inflammation [3,6,74,77].

The first-line therapy of hemochromatosis treatment is phlebotomy [70,76]. This approach involves the removal of 450–500 mL of blood once in two weeks to twice a week until ferritin reduction to required serum levels is reached. Although phlebotomy is usually well tolerated by patients, this procedure is painful, uncomfortable and disruptive to their daily routine, and some patients have poor venous access. Moreover, some side effects are often experienced after phlebotomy as a result of decreasing overall blood volume, namely tiredness [70,76]. An option to phlebotomy (in case of intolerance or contraindication) is therapy with iron chelators [70,73,76]. The traditional chelator is deferoxamine (see Figure 2 for structure), but it is poorly absorbed from the gastrointestinal tract, so therefore it must be administered intravenously or subcutaneously and has also short serum half-life requiring frequent applications. Moreover, some side effects have been noticed, including hypotension, limiting the rate of administration, ophthalmic and auditory toxicity; bacterial and fungal infections; changes in blood histology; allergic and skin reactions, especially at the site of application; and pulmonary, renal and neurological effects. Another problem with deferoxamine treatment is that it is rather expensive. Orally administrable low-molecular-weight iron chelators, such as deferiprone (Ferriprox^®^) and deferasirox (Exjade^®^) (see Figure 2 for structures), are efficient, but they also have severe side effects, such as agranulocytosis, hepatic fibrosis and renal toxicity, because they are absorbed from the gastrointestinal tract [70,73,76]. Low-iron diets have been discussed [60]; nevertheless, the important issue is avoidance of alcohol to protect oxidatively damaged liver and maintaining a broadly healthy diet.

Another case where overload occurs is as a result of transfusion therapy of thalassemia. Alfa and beta thalassemias are recessive hereditary genetic disorders associated with decreased levels of hemoglobin production leading to malfunctioning red blood cells, which do not efficiently transfer oxygen [78,79]. Thalassemias differ in severity depending on which of the hemoglobin-coding genes is the damage and how many mistakes are there. Thalassemias generally lead to anemia. Clinical manifestation varies greatly from none through mild and severe to fatal if untreated. The primary cure is regular blood transfusions, plus folate to support hemopoiesis [78,79]. However, functional blood of healthy donators used for transfusion also contains a lot of iron in its hemoglobin causing severe iron overload after regular transfusions. Thalassemia treatment, therefore, combines infusions to replenish functional blood as primary cure together with iron chelation therapy with the same chelators as in the case of hemochromatosis (deferoxamine, deferiprone and deferasirox) to suppress iron overload and shift the iron balance to normal, i.e., to suppress side effect of the primary therapy [80]. Due to obvious reasons, phlebotomy is not indicated. The simplified overview of iron metabolism, iron flow in the organism, showing points where the overload occurs due to hemochromatosis and thalassemia, and where the effect of both low-molecular-weight and polymer-drug effect takes place is shown in Figure 5.

## 4. Copper Metabolism, Diseases Connected with Its Overload and Current Treatment

Copper, another essential element, is released from the diet in acid stomach content and uptaken in the form of copper(II) by DMT1 and human copper transporter 1 (hCTR1) [81,82]. It is noteworthy that copper is secreted into the gastrointestinal tract, and further reuptake represents even more copper than what is present in food [83]. The main copper-storage organ in the body is the liver (Figure 6).

Wilson’s disease is an autosomal recessive genetic disorder caused by mutation of gene ATP7B on chromosome 13 with incidence regionally varying in the range of 1–4 cases per 100,000 (but the genetic prevalence is considerably higher, even 1:7000) [85,86,87]. Impaired function of the corresponding protein leads to high accumulation of copper in the organism, especially in the liver and central neural system, as the product of this gene is responsible for copper elimination from the liver into bile, the main regulated physiological way of excessive copper elimination from the organism. High concentrations of copper lead to typical symptoms coming from toxic oxidative damage of the liver, neural system and other organs [86]. The symptoms usually appear in teenagers as liver damage and neurological disorders. Later Wilson’s disease leads to often fatal complications as liver failure, hepatic encephalopathy and massive bleeding from esophageal varices which develop as a result of portal hypertension [85]. Liver damage usually can be at least partly restored by therapy; the progression of neurological damage can be significantly slowed down by therapy, but existing damage is typically irreversible.

Current treatment is based on lowering copper concentration in the organism by administration of copper-chelating agents [86,88,89] (penicillamine, triethylenetetramine–trientine or tetrathiomolybdate salts; see Figure 4 for structures), which lead to decreased uptake and increased elimination of copper into the urine. British anti-lewisite (dimercaprol) [90], the first agent of this type revolutionizing Wilson’s disease therapy in 1950-s, is currently obsolete, although still stockpiled chemical warfare antidote. In addition, high doses of zinc salts (even more than 1 g zinc equivalent daily, mostly as sulfate) are administrated as maintenance therapy, since they block copper uptake from the gastrointestinal tract by competitive antagonism of zinc and copper on the DMT1 [81,91]. All of these treatments, however, suffer from severe side effects [88,89], e.g., lupus and myasthenia for penicillamine, which come from the general recomplexation of essential elements within the body after uptake from the gastrointestinal tract. Severe gastrointestinal disorders and gastrointestinal irritation are frequent for zinc therapy (in fact, the doses applied cause weak zinc poisoning). The usual dietary income of copper is 0.6–1.6 mg per day, so a low-copper diet is recommended as an adjuvant measure at the beginning of therapy. Copper status is to be continuously monitored, as efficient therapy may lead to copper deficiency, also dangerous. However, copper is contained in nearly all consumables, so a copper-free diet is almost impossible [91,92]. Thus, the diet usually consists in avoiding food with high copper content (e.g., liver, mushrooms or nuts) only.

Acute high uptake of iron and copper may lead to poisoning, described from the mining industry and also for children who are susceptible, as they do not possess fully developed detoxication mechanisms. Symptoms of iron poisoning usually include gastrointestinal disorders (nausea, vomiting and diarrhea), while for copper, also jaundice and hypotension accompany gastrointestinal symptoms [23]. Therapy of acute poisoning utilizes the same chelators as therapy of hereditary diseases associated with pathological accumulation of these metals [23]. Namely, deferoxamine, deferiprone, deferasirox and clioquinol are most commonly used for acute iron poisoning [14,15,16], and penicillamine is used for acute copper poisoning [86,88,89].

## 5. Polymer Iron and Copper Chelators for Metal Overload Treatment

Polymers with bound chelators are a newer paradigm for the improvement of unfavorable pharmacokinetics and reduction of side effects of the currently used iron and copper chelators. The specific benefits of polymers depend on the way of their administration and can be roughly divided into (i) polymers to be applied parenterally and (ii) polymers to be applied orally.

### 5.1. Polymers and Nanospecies with Bound Chelators to Be Applied Parenterally

Polymers and nanospecies with bound chelators to be applied parenterally are designed to prolong the blood-circulation time of such chelators, which is especially the problem of deferoxamine. Here, the molecular weight plays the crucial role as a threshold for renal elimination, lying in the range of 6 nm diameter (corresponding to a molar mass, e.g., ca. 45 kDa for acrylamide-type polymers) [93,94]. Polymers and nanospecies (e.g., micelles, liposomes, nanogels and inorganic nanoparticles) smaller than this size may be readily eliminated by kidneys and usually possess fast elimination kinetics. Larger polymers and nanoparticles show prolonged circulation. Linear polymers usually have more diffuse renal threshold as molecules with diameter somewhat larger than the renal threshold can diffuse through the glomerular membrane via a worm-like effect in their conformations with minor probability, however much slower than the smaller molecules [95]. The prerequisite for prolonged blood-circulation time is biocompatibility.

In this way, conjugation of deferoxamine to 140 kDa polymer with Rh 10.6 nm prolonged blood-circulation time of deferoxamine 768-fold to half-life (t_1/2_) 64 h in mice [96]. Conjugation of deferoxamine to hyperbranched polyglycerol increases blood-circulation time 484-fold to t_1/2_ 44 h [97]. Polymers and nanoparticles also typically decrease the cytotoxicity of the chelators, e.g., conjugation to poly(ethylene glycol methacrylate)-based copolymers with molecular weight 27–127 kDa decreased cytotoxicity of deferoxamine more than 100-fold [98]. The reason for this effect is the limitation of biodistribution of the chelator to enter the cells when comparing low-molecular-weight and polymer-bound chelators.

Higher molecular weight and prolonged circulation, however, may lead to a redistribution of iron within the body [99], rather than elimination from the organism, although, even for non-degradable high-molecular-weight carriers, increased iron elimination was reported, perhaps by induction of hepatobiliary route of elimination [97]. Body redistribution was, for example, demonstrated on polyglycidol, where conjugation to high-molecular-weight (637 kDa) polyglycidol leads to the redistribution of iron to, e.g., liver, while lower-molecular-weight 75 kDa polyglycidol did not lead to significant organ redistribution [99]. Long polymer retention in the body is also raising the issue of polymer accumulation in the body after repeated administration with subsequent long-term toxicity. This is especially the problem of non-biodegradable polymer carriers, such as polyglycidol [99]. To solve this, biodegradable polymer carriers were proposed, such as deferoxamine-conjugated alginate cleavable by reactive oxygen species [100,101], or acid-cleavable ketal crosslinkers were employed to the polymer carriers [96]. For alginate, however, its calcium-chelating ability connected with gelation can be an issue. Analogously, deferoxamine was conjugated to enzymatically activable polyrotaxane to promote polymer carrier degradation after it fulfills its task [102,103]. Alternatively, enzymatically degradable hydroxyethylated starch can be employed [104,105].

Binding to polymer generally decreases the rate of metal chelation and especially trans-chelation from natural iron-containing proteins, such as ferritin, compared to free chelator; however, the rate is usually sufficient even for polymers or combination with low-molecular-weight chelator can be utilized [106]. This effect can be easily explained by steric reasons (trans-chelation of iron from one macromolecule to another most plausibly requires direct contact of chelating sites, as binding constants for iron are usually very high for both native biomolecule and the polymer trans-chelator) and diffusion reasons (this is especially the case of systems where the chelator is “buried” inside the polymer). Consistently with this, trans-chelation rate is less dependent on whether the trans-chelator is polymer-bound for low-molecular-weight iron-donor species. Nevertheless, polymer chelators are efficient even in vitro on cells and in vivo on animal models [96,97,98,99,100,101,102,103,104,105,106].

Chelator loading in the conjugate varies greatly in the reported systems for deferoxamine, e.g., in the range of 0.5% *w*/*w* [107] to 51% *w*/*w* [96]. The higher the chelator loading is, the lower the dosage of the conjugate is and the lower the burden for elimination is; however, higher chelator loadings may influence the biocompatibility and physicochemical characteristics of the conjugates in an unfavorable way.

Polymer architecture plays a key role in the blood-circulation time, route and ways of elimination of the carriers.

Linear polymers, such as hydroxymethylated starch [104], starch [108], poly(*N*-vinylpyrrolidone) [109] or poly(ethylene glycol methacrylate) [98], allow intermediate loadings and can be, in some cases (polysaccharides), biodegradable. They allow renal elimination even slightly above the renal threshold. If their molecular weight is sufficiently below the renal threshold, as is the case of poly(ε-lysine)-conjugated deferoxamine [110], elimination is easier; however, it also reduces blood-circulation time. If the bound chelator is bidentate, e.g., 3-hydroxypyridin-4-one, 8-hydroxyquinoline-5-sulfonic acid or hydroxamic acid, metal chelation may lead to ionic gelation and ionic crosslinking of the polymers by the chelated metal ion. If the polymer backbone is chelating and bears hydrophilic polymer biocompatible grafts, sterically stabilized defined nanoparticles may be self-assembled by iron and copper addition in this way [111,112,113,114]. Branched and hyperbranched polymers, such as dextran [104] and polyglycerol [97], are more compact and in some cases longer circulating than the linear polymers.

Dendrimers are highly defined hierarchical structures, much more regular than structurally somehow-related hyperbranched polymers, with the hydrodynamic radius usually below the renal threshold and therefore with renal elimination as the primary way of elimination. Iron-chelating dendrimers were synthesized [102,115] with deferoxamine iron-chelating moieties. Nanogels are gel-structured nanoparticles and with appropriate iron-chelating moieties [116,117,118,119]; however, due to their larger size, they must be biodegradable to allow elimination. The use of rotaxanes, where cyclodextrin rings with conjugated deferoxamine threaded on poly(ethylene oxide) axis end-capped with enzyme cleavable stoppers, is a supramolecular approach to biodegradability [103].

Another strategy is not to use a covalently bound chelator, but to modify unfavorable pharmacokinetics by sustained release of the free chelator. The advantage is that the elimination route of the chelate is essentially the same as if the chelator would be administrated without carrier; however, the polymer carrier still has to be degradable and eliminable from the organism*. I.p.* and *s.c.* applications of polymer devices releasing iron- chelating drugs deferoxamine and salicylaldehyde isonicotinoyl hydrazone in the course of seven days were described [120]. Encapsulation of deferoxamine into liposomes has a similar effect [121].

The abovementioned cases are iron sequestrants. Polymer copper chelators have been described not as excessive copper sequestrants, but as carriers of ^64^Cu, which is an excellent theranostic radionuclide for nuclear medicine that is especially useful in oncological positron emission tomography (PET) diagnostics and internal radiotherapy [122,123,124,125]. Here, the thermodynamically stable and kinetically inert complexes of radioactive copper are applied parenterally in a complexing agent with a targeting feature to the tissue of interest. Reviewing these polymer and nanoparticle ^64^Cu carriers for nuclear medicine is beyond the scope of this review and can be found elsewhere [122,123,124,125]. Moreover, chemistry applied to their construction may undoubtedly bring inspiration to the development of polymer therapeutics of copper overload.

### 5.2. Polymers with Bound Chelators to Be Applied Orally

The uptake–elimination balance of iron and copper can be shifted towards elimination by inhibiting uptake from the diet by (polymer) sequestrants. Should such a sequestrant be present in the gastrointestinal tract, it can irreversibly bind the metal after its release from the diet during the digestion process and prevent its uptake into the organism. As polymer cannot be uptaken from the gastrointestinal tract, it is eliminated with feces together with the metal chelated to it.

This approach was first developed not for the treatment of metal overload, but for sequestering bile acids to prohibit their recycling to in result reduce blood cholesterol levels as bile acids are cholesterol metabolites [126,127]. Inorganic metal-chelating materials can be also used in this way; for example, Prussian blue is stockpiled as ^137^Cs radionuclide antidote sequestrant for the case of nuclear reactor accidents [128].

Oral use of polymers brings much less safety issues than the parenteral one (see above) as polymers are inherently non-absorbable from the diet unless they are degradable to low-molecular-weight fragments. Therefore, if we assure non-biodegradability, the polymer is automatically safe from the bioaccumulation point of view. However, there is also one limitation compared to parenterally applied systems—such polymers can be used to treat diseases where the metal input is from the diet (e.g., Wilson’s disease, hemochromatosis or copper/iron acute oral poisoning), but they are inefficient to treat blood infusion-induced iron overload occurring during the therapy of thalassemia where the iron input is not from diet. There is also an unanswered question of how such chelating polymers would influence gut microbiota, as copper and especially iron chelators possess antimicrobial action by depleting these essential metal micronutrients to bacteria. For example, bacteriostatic effects under in vitro conditions on several opportunistic pathogens have been clearly demonstrated for polymer-bound iron chelators [109]. However, especially in vivo data are missing.

The design of polymers for oral use is quite different from the design of parenteral systems. Linear polymers [129] and dendrimers [130] have also been described; however, most described systems for this use are crosslinked polymer beads and hydrogel microparticles. Microparticles with limited diffusion of proteins into their structures possess the advantages of reducing unwanted interactions with the highly complex environment of the intestinal content (partly digested diet, as well as gastrointestinal tract mucosa). Interaction with a partly digested diet is typical for plant polyphenols, namely for tannins. Tannins are strong chelators of iron(III) and other polyvalent ions present in a vegetarian diet (e.g., for these reasons, tannin-rich tea is recommended for hemochromatosis patients) [60]. However, a high-protein diet causes the formation of interpolyelectrolyte complex precipitates between anionic tannins and proteins and their fragments which can be cationic [131,132]. These complexes do not possess the astringent bitter taste of tannins (e.g., this is why milk is sometimes added to coffee and tea to make them tastier), but they also do not efficiently complex metal ions. In fact, gelatin tannate is used as an antidiarrheal with proven non-interference with metal nutrients uptake [133]. Most oral polymer iron scavengers under development (see below) are phenolics, so this is a major issue. The interaction with the mucosa of the gastrointestinal tract may be beneficial to form a temporary barrier depot increasing efficacy of the treatment [134], but may potentially disturb the digestion process in long-term effects; however, this was not shown in (mostly short term) animal studies. 

The polymer architecture is also of great importance. Hydrogel structures [84,135,136,137,138] limit diffusion of higher-molecular-weight substances into the microparticle and provide conformational freedom to the chelating ligands, which is of special advantage for bidentate ligands that readily form complexes with more than one ligand per metal ion stoichiometry [112,113]. The same was seen for micellar systems [112,113,139]. Ion imprinting can further facilitate pre-adjustment of the ligands to appropriate geometry [117]. However, the introduction of hydrophobic ligands may together with this additional crosslinking lead to limitation in metal binding capacity [137]. Macroreticular resin microparticles [134,140,141] with double molecular porosity (gaining access to the chelated metal to the moieties) and meso/macropores (accelerating strongly metal chelation kinetics) have the main advantage of fast metal uptake kinetics, usually within a few minutes; metal is completely complexed from solution (see Figure 6, for example), even for larger particles, where diffusion would otherwise slow down the uptake.

However, the trade-off is lower chelation capacity as considerable weight fraction of the macroreticular structure is non-chelating crosslinker.

Material chemistry of the polymer support of the microparticle also matters. It must be hydrophilic to allow swelling in aqueous media, providing accessibility of metal to the ligands as some metal chelating ligands are hydrophobic. Overall charge also matters, polycationic carriers (e.g., polyethyleneimine, polyallylamine and chitosan) [135,137,142,143] may Coulombically repulse metal cations, thus decreasing overall chelating capacity, while polyanionic (e.g., carboxymethyl cellulose and alginate) [100,143] may also nonspecifically bind other polyvalent cations, such as calcium or zinc.

The chelating iron ligands used for microparticles are mostly inspired by bacterial siderophores such as deferoxamines and enterobactin or plant phenolics, e.g., catechol, caffeic acid, 2,5-dihydroxybenzoic acid, [134,135] 3-hydroxypyridin-4-ones, [111,130,143,144,145] 2,3-dihydroxybenzoic acid [137] or hydroxamic acid [136]. These groups efficiently complex iron(III) which is the most abundant form of iron in the diet, but not iron(II). As the reduction of iron(III) to iron(II) is the prerequisite for iron uptake (Fe(III) as-is is not uptaken), binding also iron(II) brings additional benefits. Our group successfully utilized polymer-bound 9,10-phenantroline, as it readily complexes both iron (II) and iron (III) for this purpose [134].

Copper chelation ligands involved are mainly inspired by those used for the treatment of Wilson’s disease in low molecular form (triethylenetetramine in our studies), [84,140,141] for copper removal from wastewater (dipicolylamine or hydroxamic acid in our studies) [84,112,140,141] or by cloroxine, antibacterial agent widely used for the treatment of diarrhea (conjugated by Mannich reaction, in non-sulfonated of sulfonated form, in our studies) (see Figure 7 for structures) [84,113,140,141].

Strong phosphate–iron(III) binding is of great importance for plant nutrition in agriculture, but as it is not completely specific for iron, it is of limited use in medicine. From the dietary point of view, phytin, a micro/nanoparticulate *myo*-inositol-1,2,3,4,5,6-hexakisphosphate calcium/magnesium/zinc complex, ubiquitous in plant material, greatly decreases the bioavailability of iron (but also of zinc, magnesium and calcium) from a vegetarian diet. Phytin also has a strong influence on gut microbiota (see References [60,146,147,148].

Another approach to prolonging the action of orally applied chelating therapeutics is a sustained release oral formulation, as demonstrated on deferiprone [149].

The approaches are summarized in Table 2.

## 6. Conclusions and Further Challenges in Polymer Iron- and Copper-Chelating Therapeutics

Although efficient treatment of iron and copper overload with low-molecular-weight chelators has already existed for decades, serious side effects, often unfavorable pharmacokinetics and limited efficacy of these agents force the development of new avenues to more safe and efficient therapeutics. Polymers hold great promise to become therapeutics of the future; however, there are questions to be answered and challenges to be overcome.

For the therapy of iron overload, most polymers described so far chelate just iron(III), which is most abundant in the diet, but also the least absorbable. Its uptake requires reduction to Fe(II) following transport by DMT1 into cells of the intestinal mucosa, and both of these steps can be inhibited. There exist experimental low-molecular-weight inhibitors for both these steps (e.g., hexacyanoferrate for iron(III) reduction [150] and bis-thiuronium salts for DMT1 inhibition [151]), but polymer inhibitors with these moieties bound to a macromolecular carrier may be beneficial and synergic with iron(III) chelation and may utilize the advantage of macromolecules of reducing side effects by keeping the effect to be limited to intestinal content. To the best of our knowledge, no such polymers have been described so far. In fact, DMT1 transporter inhibitors may be beneficial not only for treatment of iron overload, but also for treatment of Wilson’s disease (copper(II) is also uptaken by DMT1 transporter; therapy with zinc(II) salts is based on this fact) and lead poisoning. There are also only rare cases of polymers complexing iron(II) [134].

Another challenge is hem chelation as heme-bound iron is uptaken in heme complex with a separate transporter and decomplexed inside cells only after uptake; here, polymer-bound heme-complexing agents, such as chloroquine, may find use (chloroquine is an antimalarial agent antiparasitic effect of which is strongly believed to be due to heme complexing in plasmodium-infected erythrocytes).

Another unanswered challenge is long-circulating injectable polymer copper(II) chelators as Wilson’s disease therapeutics.

Iron- and copper-chelating polymers may also find use not only as metal overload therapeutics but also in other medical applications—as antibacterial agents (as copper and especially iron [109] are essential and growth-limiting for bacteria) [18,152], as anticancer agents [153,154] (essentiality of iron and copper is especially pronounced in quickly growing cells) and treatment of inflammation-caused blood clots after COVID-19 infection and brain injuries (here, fibrin clots are reinforced and made less biodegradable by reactive oxygen species, for which the production of iron is essential) [155,156,157]. The ROS-generating metals play a key role in the development of Alzheimer’s disease by direct interaction with the beta amyloid [18,21,89], as well as in the development of some other neurodegenerative diseases.

## Figures and Tables

**Figure 1 polymers-13-03969-f001:**
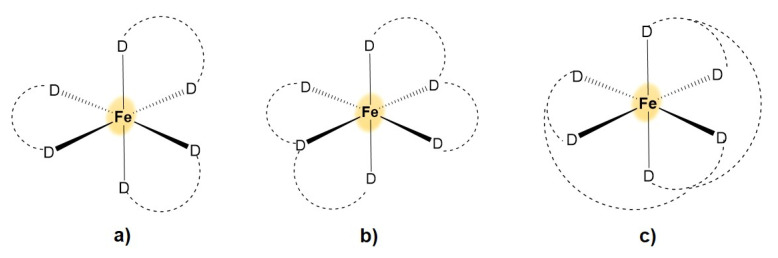
Octahedral complexation geometry of Fe(III) obtained with ligands of different denticities: (**a**) bi-, (**b**) tri- and (**c**) hexadentate ligands.

**Figure 2 polymers-13-03969-f002:**
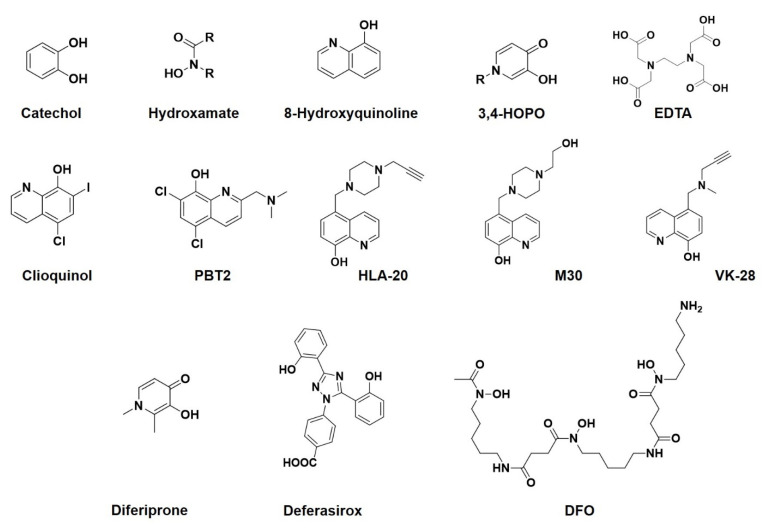
Selected iron (III) chelators.

**Figure 3 polymers-13-03969-f003:**
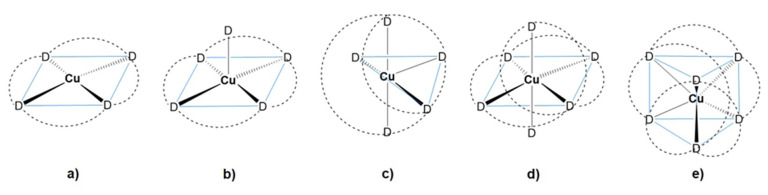
Complexation geometries of Cu(II) complexes: (**a**) square-planar, (**b**) square-pyramidal, (**c**) trigonal-bipyramidal, (**d**) octahedral and (**e**) trigonal-prismatic.

**Figure 4 polymers-13-03969-f004:**
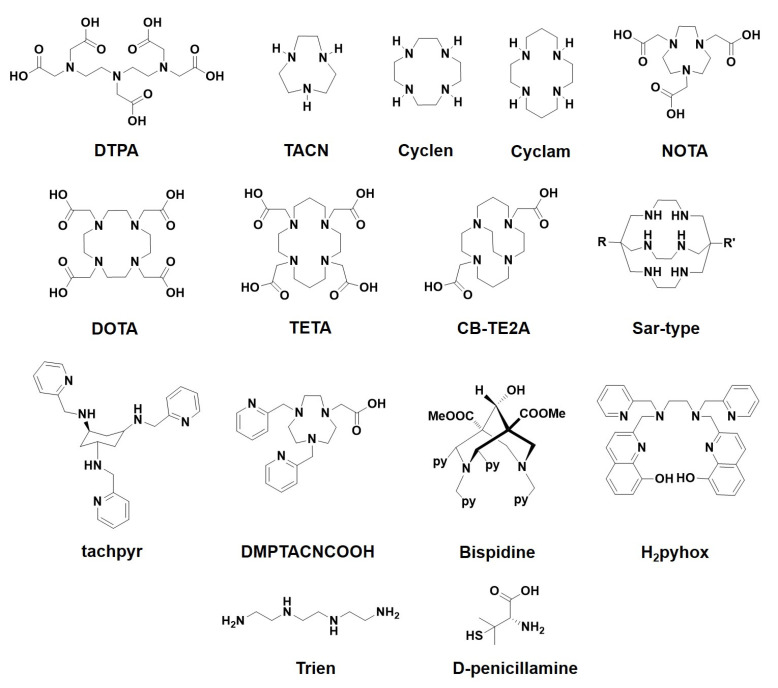
Selected copper (II) chelators.

**Figure 5 polymers-13-03969-f005:**
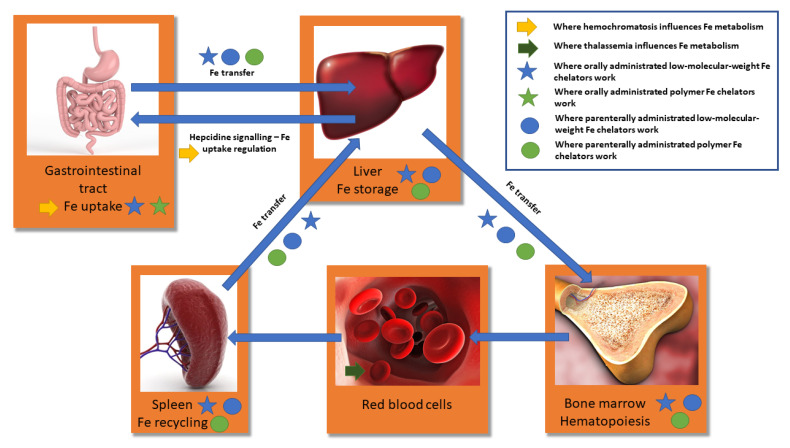
Simplified overview of iron metabolism and iron flow in organism shows where the overload occurs due to hemochromatosis and thalassemia and where the effect of both low-molecular-weight and polymer-drug effect takes place.

**Figure 6 polymers-13-03969-f006:**
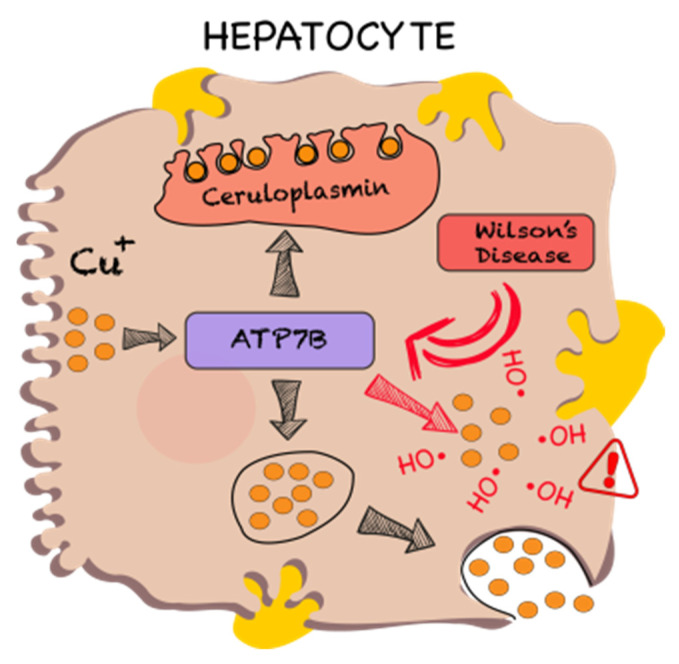
Mechanism of copper accumulation in hepatocyte. Reprinted with permission from Reference [84]. Copyright 2021 Elsevier.

**Figure 7 polymers-13-03969-f007:**
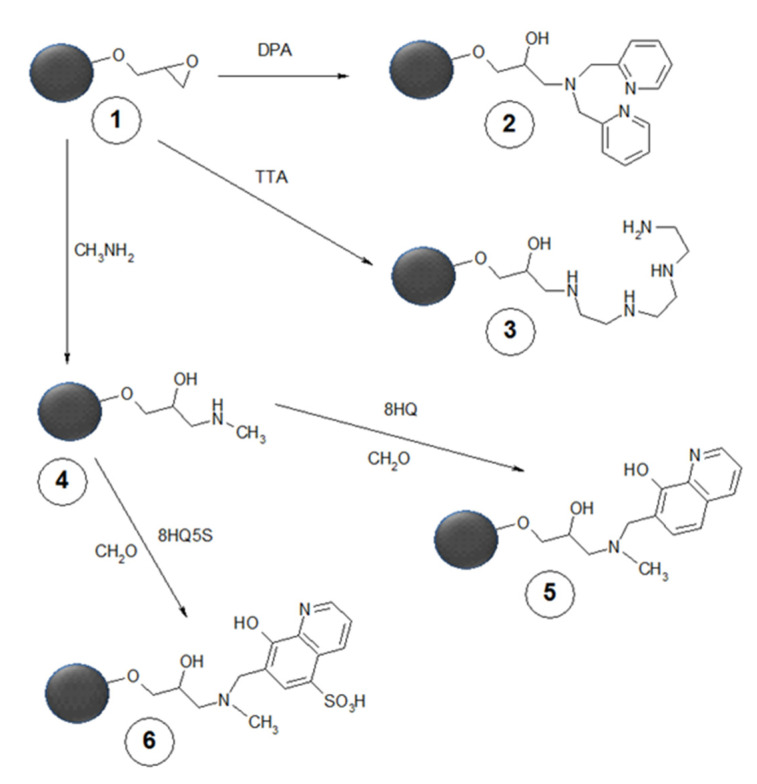
Macroreticular copper scavengers (2,3,5,6) for oral use synthesized from poly(glycidyl methacrylate-co-ethylene methacrylate) (1); the Mannich condensation requires secondary amine intermediate 4. Reprinted from Reference [141], with permission. Copyright 2021 Elsevier.

**Table 1 polymers-13-03969-t001:** Summary of most relevant iron and copper chelator types.

**Important Iron Chelator Types**			
**Chemistry**	**Donor Atoms**	**Examples**	**References**
Hydroxamate type	N,O	DFO	[28]
Hydroxypyridone type	O,O	Diferiprone, DIBI, 3,4-HOPO	[14,24,25]
8-Hydroxyquinoline type	N,O	Clioquinol, PBT2, HLA20, M30, VK-28	[18,19,20,21,22,23]
1,2,4-Triazol type	N,O	Deferasirox	[26,27]
**Important Copper Chelator Types**			
**Chemistry**	**Donor Atoms**	**Examples**	**References**
Open-chain polyamines	N	Trien	[9,23,56,57]
Open-chain polyaminocarboxylates	N,O	EDTA, DTPA	[31,32]
Azamacrocycles	N	TACN, cyclen, cyclam	[33,34,35]
Macrocyclic polyaminocarboxylates	N,O	DOTA, NOTA, TETA	[36,37,38]
Caged azamacrocyclic systems	N	CB-TE2A, Sarcophagine	[39,40,48,49]
Pyridine ligands	N	tachpyr, dipicolylamine, DMPTACNCOOH, bispydine	[41,42,43,44,45,46,47,50,51,52,53,54]
8-Hydroxyquinoline type	N,O	H2Pyhox	[55]
Thiol-amine type	S,N	D-penicillamine	[58,59]
Tetrathiomolybdate	S	Tetrathiomolybdate	[9,23,58,59]

**Table 2 polymers-13-03969-t002:** Summary of polymer chelator approaches.

Administration Route	System Architecture	Biodegradable	References
Parenteral	Linear	No	[98]
Parenteral	Linear	Yes	[100,101,104,105,109,110]
Parenteral	Hyperbranched	No	[97,99]
Parenteral	Crosslinked	Yes	[96]
Parenteral	Polyrotaxane	Yes	[102,103]
Parenteral	Nanogel	No	[111,112,113,114,116,117,118,119]
Parenteral	Dendrimer	No	[106,115]
Sustained release of free drug	Device	No	[120]
Sustained release of free drug	Liposome	Yes	[121]
Oral	Linear	No	[129]
Oral	Dendrimer	No	[130]
Oral	Hydrogel	No	[84,135,136,137,138]
Oral	Micelles	Yes	[112,113,139]
Oral	Crosslinked, imprinted	No	[117]
Oral	Macroreticular	No	[84,134,140,141]
Sustained release of free drug	Device	No	[149]

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
