# Peer review of "Chelators for Treatment of Iron and Copper Overload: Shift from Low-Molecular-Weight Compounds to Polymers"

_polymers, 2021, doi:10.3390/polym13223969_

Round 1
Reviewer 1 Report
The work summarizes the chemical properties for efficient chelation of the common forms of iron and copper ions, the metabolism of iron and copper, and the diseases associated with their overload. More original is the review of the different approaches to modify existing chelators to increase their molecular weight and pharmacokinetics. The work is well presented and organized, but some weaknesses can be improved.
- A table to summarize the different approaches for polymeric chelators would facilitate the reading.
- Many different approaches to increase the molecular weight of the chelators are shown, and their effects on metal chelation and pharmacokinetics are discussed. What is missing is the indication of their efficacy in the removal of iron in animals and cells. An obvious curiosity of the reader is how these modified desferrioxamine's compare with the unmodified ones in removing the iron burden.
- regarding the oral polymeric chelators, the authors may add information on how they affect the gut microbiota.
- The sections dedicated to copper are minor compared with those to iron, but 5 figures are for copper and two for iron. A better balance is needed. The importance of figure 6 for the review seems minor.
- ref 75. The author list is far too long, as it occupies 2 full pages.
Author Response
Dear Editor,
Thank you very much for valuable comments to our manuscript # [Polymers] Manuscript ID: polymers-1438959. After checking English we have carefully point-by-point replied to all objections of both reviewers as follows (all changes have been made in track changes mode):
Response to reviewers` objections
Reviewer 1
Objection: The work summarizes the chemical properties for efficient chelation of the common forms of iron and copper ions, the metabolism of iron and copper, and the diseases associated with their overload. More original is the review of the different approaches to modify existing chelators to increase their molecular weight and pharmacokinetics. The work is well presented and organized, but some weaknesses can be improved.
Our reply: Thank you very much for positive comments on our manuscript.
Objection: - A table to summarize the different approaches for polymeric chelators would facilitate the reading.
Our reply: The table suggested was added (now Table 2 in the revised manuscript).
Objection: - Many different approaches to increase the molecular weight of the chelators are shown, and their effects on metal chelation and pharmacokinetics are discussed. What is missing is the indication of their efficacy in the removal of iron in animals and cells. An obvious curiosity of the reader is how these modified desferrioxamine's compare with the unmodified ones in removing the iron burden.
Our reply: The following discussion was added to the manuscript, section 5.1. Polymers and nanospecies with bound chelators to be applied parenterally: ” Binding to polymer generally decreases the rate of metal chelation and especially trans-chelation from natural iron-containing proteins such as ferritin compared to free chelator; however, the rate is usually sufficient even for polymers or combination with low-molecular-weight chelator can be utilized. [108] This effect can be easily explained by steric reasons (trans-chelation of iron from one macromolecule to another most plausibly requires direct contact of chelating sites as binding constants for iron are usually very high for both native biomolecule and the polymer trans-chelator) and diffusion reasons (this is especially the case of systems where the chelator is “buried” inside the polymer). Consistently with this, trans-chelation rate is less dependent on whether the trans-chelator is polymer-bound for low-molecular-weight iron donor species. Nevertheless, polymer chelators are efficient even in vitro on cells and in vivo on animal models.[98-108]”
Objection: - regarding the oral polymeric chelators, the authors may add information on how they affect the gut microbiota.
Our reply: The discussion concerning effect of polymer chelators to gut microbiota was added to section 5.2 „Polymers with bound chelators to be applied orally“. We also added a discussion about a prominent effect of nanoparticulate/colloidal iron chelator phytin, omnipresent in diet of plant origin, on intestinal bacteria.
Objection: - The sections dedicated to copper are minor compared with those to iron, but 5 figures are for copper and two for iron. A better balance is needed. The importance of figure 6 for the review seems minor.
Our reply: The purpose of the review is to highlight further development based on the published data and while the macromolecular approach is already somehow established for iron, but for copper i tis now a hot topic, so we wish to highlight it. But we agree that the balance should be more kept and therefore we omitted Figure 6 concerning copper. We also omitted some redundant discussion concerning iron overload-related diseses haemochromatosis and thalassemia (philosophical discussion what is disease).
Objection: - ref 75. The author list is far too long, as it occupies 2 full pages.
Our reply: We have shortened extensive list of co-authors of ref. 75 by stating the first five and then „et.al.“.
We now feel our manuscript is significantly improved.
Best regards
Martin Hrubý

Reviewer 2 Report
The review presented by Hruby and colleagues discusses recent changes within the field of iron and copper chelation. They present critical background information including the physiological roles of iron and copper, and the coordination chemistry associated with the metals. This information leaves the reader well-situated to discuss iron and copper metabolism and the potential treatment strategies for the management of overload. The main focus of the review is the use of polymers, rather than low-molecular-weight compounds, for iron and copper chelation. The authors illustrate the utility of these polymers, dividing the compounds into two categories based on route of administration for further discussion.
Overall, this review represents a valuable addition to the literature and can serve as a reference paper for scientists within the field. While many reviews exist that cover chelation as a therapeutic strategy, this review fills a gap within the literature by comparing low-molecular-weight compounds and polymers for both copper and iron chelation. The review is quite comprehensive and provides a good overview of where the field has been and where it is heading. In general, it is well-written and makes good use of figures.
Specific comments:
Introduction: While the liver is certainly affected, other organs can also be damaged. It would be useful to elaborate on this a bit and include a citation to allow the reader to access more detailed information if required. It would also help to expand on the negative effects seen with the low-molecular-weight chelators to help frame the issue.
Line 51: You indicate that there are clear differences between copper and iron, but the specifics of copper are not given at this point. It would be helpful to include this information early into this section to allow for the comparison of the basic characteristics of the two metals. You can then provide further detail for each individual metal.
Line 82: Why does low selectivity for iron lead to toxicity? You could illustrate this with a specific example of one of these compounds leading to in vivo toxicity.
Line 127: Listing the names of the various copper chelators creates a fairly dense wall of text. Perhaps these could be provided in a table. One suggestion would be to create a table outlining both the copper and iron chelators.
Figure 5: It would be beneficial to have a figure for iron metabolism as well.
Line 255: While interesting, this content does not fit with the topic of the review.
Line 308: Acute poisoning is an interesting topic, but for inclusion in this review it should be fleshed out a bit with some additional details and references.
Section 5.1: This paragraph was a bit unclear at times and could use some restructuring to add better balance to the section. For example, the first paragraph is quite long, while the next few are very short. One suggestion is to separate the concepts of elimination and redistribution.
DIBI is a novel iron-binding polymer that could be mentioned in this section. Please see the following reference.
Ang, M., Gumbau-Brisa, R., Allan, D., McDonald, R., Ferguson, M., Holbein, B., & Bierenstiel, M. (2018). DIBI, a 3-hydroxypyridin-4-one chelator iron-binding polymer with enhanced antimicrobial activity. MedChemComm, 9(7), 1206-1212.
Line 398: While I agree that nuclear medicine is beyond the scope of this review, this section could be expanded slightly. Can you speculate on possible future applications?
Definitions:
There are a couple of instances in the review where some definitions could be given.
Line 55: Please ensure the definition of HSAB is clear.
Line 103: Please expand on what you mean by a “borderline” element as not all readers will be familiar with HSAB theory.
Line 321: Nanospecies should be defined.
Author Response
Dear Editor,
Thank you very much for valuable comments to our manuscript # [Polymers] Manuscript ID: polymers-1438959. After checking English we have carefully point-by-point replied to all objections of both reviewers as follows (all changes have been made in track changes mode):
Response to reviewers` objections
Reviewer 2
Objection: The review presented by Hruby and colleagues discusses recent changes within the field of iron and copper chelation. They present critical background information including the physiological roles of iron and copper, and the coordination chemistry associated with the metals. This information leaves the reader well-situated to discuss iron and copper metabolism and the potential treatment strategies for the management of overload. The main focus of the review is the use of polymers, rather than low-molecular-weight compounds, for iron and copper chelation. The authors illustrate the utility of these polymers, dividing the compounds into two categories based on route of administration for further discussion.
Overall, this review represents a valuable addition to the literature and can serve as a reference paper for scientists within the field. While many reviews exist that cover chelation as a therapeutic strategy, this review fills a gap within the literature by comparing low-molecular-weight compounds and polymers for both copper and iron chelation. The review is quite comprehensive and provides a good overview of where the field has been and where it is heading. In general, it is well-written and makes good use of figures.
Our reply: Thank you very much for positive comments on our manuscript.
Objection: Specific comments: Introduction: While the liver is certainly affected, other organs can also be damaged. It would be useful to elaborate on this a bit and include a citation to allow the reader to access more detailed information if required. It would also help to expand on the negative effects seen with the low-molecular-weight chelators to help frame the issue.
Our reply: The examples of other organs affected and side effects of the current treatment were added to Introduction with remark that details are discussed in detail later in Chapters 3 and 4.
Objection: Line 51: You indicate that there are clear differences between copper and iron, but the specifics of copper are not given at this point. It would be helpful to include this information early into this section to allow for the comparison of the basic characteristics of the two metals. You can then provide further detail for each individual metal.
Our reply: The explanation „…, mainly coming from different preferred coordination geometry, size, charge and polarizability of their ions …“ was added to the specified place.
Objection: Line 82: Why does low selectivity for iron lead to toxicity? You could illustrate this with a specific example of one of these compounds leading to in vivo toxicity.
Our reply: The explanation was extended on the specified place providing also examples: „However, due to a rather low chelation selectivity of these 8-hydroxyquinoline derivatives (clioquinol, PBT2, HLA20, M30 and VK-28) for iron(III) ions, these derivatives may exhibit in vivo toxicity. This is because they may chelate, e.g., copper(II) or zinc(II) alongside with iron(III) causing depletion of these essential micronutrients further leading to inhibition of metalloenzymes dependent on them.[23]“
Objection: Line 127: Listing the names of the various copper chelators creates a fairly dense wall of text. Perhaps these could be provided in a table. One suggestion would be to create a table outlining both the copper and iron chelators.
Our reply: The table summarizing the most important iron and copper chelator types was added to the manuscript (now Table 1 in the revised manuscript)
Objection: Figure 5: It would be beneficial to have a figure for iron metabolism as well.
Our reply: The figure describing simplified iron metabolism, iron flow in organism, showing where the overload occurs due to hemochromatosis and thalassemia and where the effect of both low-molecular-weight and polymer-drug effect takes place was added to the manuscript (now Figure 5).
Objection: Line 255: While interesting, this content does not fit with the topic of the review.
Our reply: The philosophical discussion about what is disease and what is advantage was omitted.
Objection: Line 308: Acute poisoning is an interesting topic, but for inclusion in this review it should be fleshed out a bit with some additional details and references.
Our reply: The discussion about acute iron and copper poisoning including symptoms and therapy was extended in the specified place including addition of more references.
Objection: Section 5.1: This paragraph was a bit unclear at times and could use some restructuring to add better balance to the section. For example, the first paragraph is quite long, while the next few are very short. One suggestion is to separate the concepts of elimination and redistribution.
Our reply: The first paragraph of section „5.1. Polymers and nanospecies with bound chelators to be applied parenterally“ was split to 3 paragraphs to make the text clearer: The first paragraph with general rules, the second with circulation and the third with redistribution/elimination.
Objection: DIBI is a novel iron-binding polymer that could be mentioned in this section. Please see the following reference. Ang, M., Gumbau-Brisa, R., Allan, D., McDonald, R., Ferguson, M., Holbein, B., & Bierenstiel, M. (2018). DIBI, a 3-hydroxypyridin-4-one chelator iron-binding polymer with enhanced antimicrobial activity. MedChemComm, 9(7), 1206-1212.
Our reply: The suggested reference was added to the Sections 5.1 („Polymers and nanospecies with bound chelators to be applied parenterally“), 5.2 („Polymers with bound chelators to be applied orally“) and 6 („Conclusions and further challenges in polymer iron and copper-chelating therapeutics“) with relevant discussion as suggested.
Objection: Line 398: While I agree that nuclear medicine is beyond the scope of this review, this section could be expanded slightly. Can you speculate on possible future applications?
Our reply: The paragraph about nuclear medicine was extended, especially discussing potential of 64-Cu carriers designed for nuclear medicine to inspire the further development of paretneral copper chelators as therapeutics of coper overload.
Objection: Definitions: There are a couple of instances in the review where some definitions could be given. Line 55: Please ensure the definition of HSAB is clear.
Our reply: A definition and explanation of the HSAB concept was added to the section 2. „Coordination chemistry of iron and copper“: „Iron ions occur in biological systems mainly in the oxidation states 2+ (d6 configuration) and 3+ (d5), which are stabilised by different ligands according to Pearson's hard and soft (Lewis) acids and bases (HSAB) concept of hard and soft acids and bases. [12] This concept deems species that are small, highly charged and weakly polarizable as “hard” and species that are large, lowly charged and highly polarizable as “soft”. “Hard” Lewis acids prefer “hard” Lewis bases and vice versa.“
Objection: Line 103: Please expand on what you mean by a “borderline” element as not all readers will be familiar with HSAB theory.
Our reply: The explanation was added to the sentence as follows: „The Cu(II) can be classified as a "borderline" (something between “hard” and “soft” Lewis acid) element according to Pearson's HSAB concept.“. The HSAB concept is explained above at the beginning of the chapter.
Objection: Line 321: Nanospecies should be defined.
Our reply: The most important examples of nanospepcies were added to the section 5.1 „Polymers and nanospecies with bound chelators to be applied parenterally“: „…(e.g., micelles, liposomes, nanogels, inorganic nanoparticles).“
We now feel our manuscript is significantly improved.
Best regards
Martin Hrubý

Round 2
Reviewer 1 Report
the suggestions of the reviewers have been accepted and the manuscript improved.
Reviewer 2 Report
Thank you very much for the revision of the manuscript!